# Internal plication for spring confinement to lengthen intestine in a porcine model

Talha A. Rafeeqi[1�¤], Anne-Laure Thomas[1�¤], Fereshteh Salimi-Jazi[1�¤], Modupeola Diyaolu[1�¤], James C. Y. Dunn[1,2¤]*

1 Division of Pediatric Surgery, Department of Surgery, Stanford University, Stanford, CA, United States of America, 2 Department of Bioengineering, Stanford University, Stanford, CA, United States of America

These authors contributed equally to this work.
¤ Current address: Division of Pediatric Surgery, Center for Academic Medicine, Department of Surgery, Stanford University, Stanford CA, United States of America
* jdunn2@stanford.edu

## Abstract

### Background

Short bowel syndrome and its resultant nutritional deficiencies are the most common cause of intestinal failure. Significant intestinal lengthening using intraluminal springs is feasible in porcine models using an external plication technique. We hypothesize that an internal plication technique will yield significant intestinal lengthening, which may lead to future endoscopic spring placement.

### Methods

Uncompressed springs measuring 7.5 cm with a diameter of 1.0 cm were compressed to 2.0 cm. A gelatin-encapsulated compressed nitinol spring was inserted into the jejunal lumen of juvenile pigs and held in place with endoluminal sutures just proximal and distal to the spring-containing segment. A control segment distal to the spring was marked. Pigs were euthanized on postoperative day 7. Spring and control segments were collected for analyses.

### Results

There was an average lengthening by 72% of the spring segment compared to the control segment. Two out of 7 springs stayed within both sets of plications and doubled in length. Histology showed normal mucosal integrity of the spring segment and plicated areas with similar muscular thickness but increased crypt depth and villus length compared to the control segment.

### Conclusion

Internal plication resulted in significant bowel lengthening. Five springs had slipped through proximal, distal or both sets of plications, resulting in less lengthening than those that remained fixed. A more consistent methodology for endoluminal suturing is needed to produce more lengthening.

**Data Availability Statement:** All relevant data are within the paper and its Supporting Information files.

**Funding:** This work was supported by the National Institutes of Health [R01DK130972] awarded to

James C.Y. Dunn, MD PhD. The NIH did not play a role in study design, data collection and analysis, decision to publish or preparation of the manuscript. https://reporter.nih.gov/search/Lw0mL7cBJk-dGo_NlVYp_A/project-details/10338557.

**Competing interests:** I have read the journal's policy and the authors of this manuscript have the following competing interests: James CY Dunn, MD PhD is a co-founder of Eclipse Regenesis, a company that develops similar spring-based intestinal lengthening devices, and reported his patent on "Expandable distension device for hollow organ growth. This does not alter our adherence to PLOS ONE policies on sharing data and materials.

## Introduction

Short bowel syndrome is a devastating gastrointestinal disease that is commonly caused by intestinal atresia, gastroschisis, and necrotizing enterocolitis in children [1]. These diseases as well as their surgical treatments can result in decreased functional intestine, hindering mucosal absorptive capacity and resulting in intestinal failure (IF). IF may result in additional comorbidities such as failure to thrive, sepsis and mortality [1]. Treatment is primarily medical, involving expensive long-term parenteral nutrition and its potential complications such as central line infections and liver failure [2,3]. Intestinal lengthening operations have been utilized but require the presence of dilated intestine and are associated with their own morbidities such as hemiloop ischemia in the Bianchi procedure and redilation with the need for reoperation in the STEP procedure [4].

Our laboratory has been experimenting with spring-mediated distraction as a form of intestinal lengthening without intestinal division. We have previously demonstrated significant intestinal lengthening of the jejunum and colon in murine and porcine models [5–10]. In those experiments, a gelatin-encapsulated spring was implanted into a segment of bowel via an enterotomy at laparotomy. The spring was affixed with externally plicating sutures placed in the bowel wall immediately proximal and distal to the compressed spring. By the time the animals were euthanized, spring-containing intestinal segments had lengthened up to 3-fold [5,11]. We have also shown that multiple springs can be placed in continuity with minimal adverse effects, thus resulting in an overall more substantial intestinal lengthening relative to individual segments [5]. Though these findings have not yet been translated to clinical applications, they still represent an innovative treatment option by directly addressing the shortened functional gut that causes IF.

Our ultimate goal is to develop a minimally invasive method of spring implantation to treat IF. The main steps of spring implantation consist of spring delivery and spring fixation. Spring fixation is currently invasive as it necessitates direct bowel manipulation to place plicating sutures from the serosal side of the intestine. As a first step toward developing an endoscopic method to lengthen the intestine, we hypothesized that internal plication of the lumen to confine springs would lead to significant jejunal lengthening if the springs were held by the luminally placed sutures.

## Methods

### Spring construction

Springs were formed by shape-setting heat treatment of coiled nitinol filaments (nickel titanium alloy, McMaster-Carr, Santa Fe Springs, CA) as described previously [5]. Active coils were calculated based on wire gauge and diameter to target a spring constant of 5 N/m [12,13]. Uncompressed springs of 7.5 cm length and outer diameter of 10 mm were constructed with 0.015" gauge nitinol wire wound around grooved steel rods and secured with hose-clamps. The springs were subjected to a heating-cooling cycle at 500˚C for 30 minutes before cooling with ice water. Spring force was measured, and the springs were then compressed to 2 cm within a gelatin capsule before they were coated with cellulose phthalate acetate and sterilized with ultraviolet (UV) light prior to implantation (**Fig 1**).

### Surgical implantation

All animal procedures were approved by Stanford University Institutional Care and Use Committee (Administrative Panel on Laboratory Animal Care Protocol #32278). Juvenile female Yucatan pigs (n = 8) were anesthetized with isoflurane and endotracheally intubated.

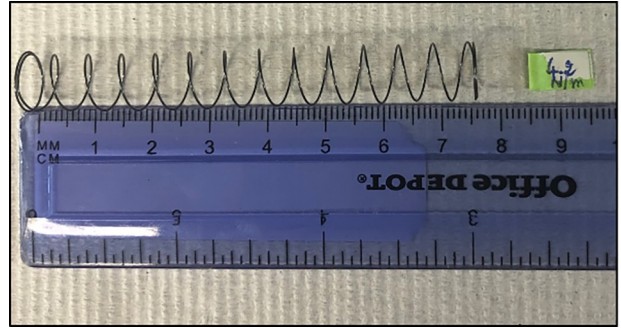
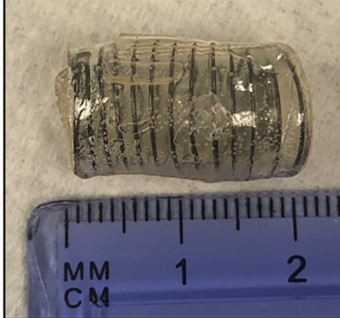

**Fig 1.** Nitinol spring prior to implantation (A) Uncompressed nitinol spring. (B) Same spring compressed within gelatin capsule prior to implantation.

Laparotomy was performed with standard aseptic technique. Small bowel was exteriorized and followed proximally to the ligament of Treitz. Jejunal transection was made at 60 cm distal to this, and the proximal end of the transected jejunum was everted using tissue forceps to expose 10 cm of the mucosal surface after measuring and marking a 2.5-cm target segment. Four submucosal 4–0 chromic gut sutures were placed in a longitudinal row to plicate the intestinal lumen to achieve 50% narrowing. A cellulose-coated compressed nitinol spring was placed next to the endoluminal sutures, and a portion of the everted bowel was reduced over the gelatin capsule, followed by the placement of another four submucosal 4–0 chromic sutures distally to affix the spring within the lumen. The rest of the everted bowel was reduced completely, and the length of the spring segment was marked with 4–0 polypropylene sutures on the serosal side. Interrupted 4–0 polypropylene sutures were used to restore jejunal continuity, and a separate control segment was marked 10 cm distal to the enterotomy with 4–0 polypropylene sutures (Fig 2). The bowel was returned to the abdomen and warm normal saline was used to irrigate the cavity. The fascia was closed in a running fashion and the skin was stapled. Pigs were fed a liquid diet to reduce the possibility of obstruction for 7 days. Pigs were euthanized on postoperative day (POD) 7.

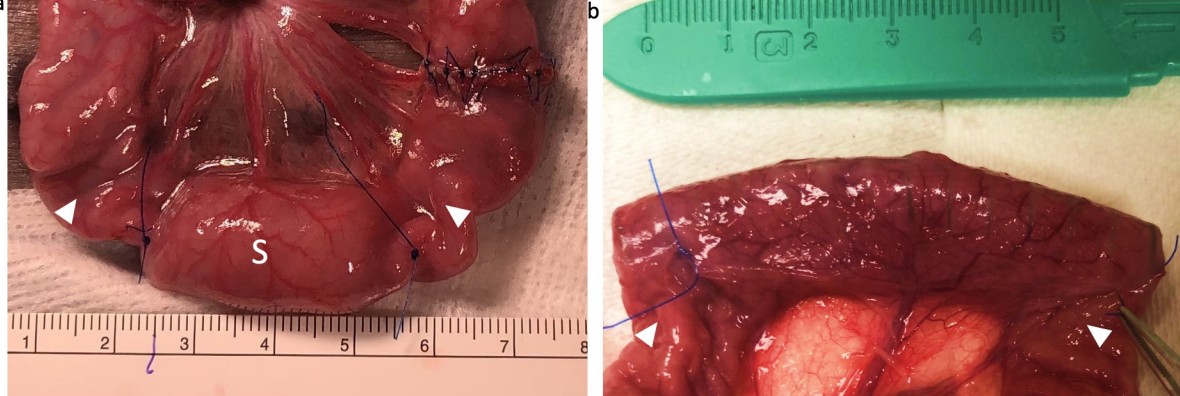

**Fig 2. Spring contained within jejunal lumen with internal plications.** Arrowheads point to internal plication sites. S denotes spring-containing segment. (A) Compressed spring within gelatin capsule placed during index operation. Note that length at time of measurement was 2.5 cm but changed due to bowel elasticity and motion at the time of measurement (B) Expanded spring contained within plications on POD 7 with segmental lengthening.

### Jejunal lengthening

After euthanasia on POD 7, spring and control segments were measured prior to retrieval. Spring and control segment lengths at time of implantation and on POD 7 were compared to quantify absolute growth of the spring segment as well as spring segment growth relative to changes in length of the control segment.

### Histopathology

Tissue specimens collected from control jejunum and spring segments were fixed in 10% formalin overnight and embedded in paraffin, aligned in perpendicular cross sections. The tissue blocks were then cut into 5-μm sections and stained with hematoxylin and eosin. Sections were examined at 12.5x magnification using light microscopy (Olympus Corporation, Waltham, MA). Representative samples were selected for each pig for both control and spring segments. Muscularis thickness, crypt depth and villus length were measured for each sample at nine different points.

### Statistical analysis

Absolute spring segment lengthening POD 7 was calculated as percentage of initial length at the time of implantation, and spring segment length was also compared to changes in control segment length. These values were averaged and reported as mean ± standard deviation. Two-tailed paired t-test was used to evaluate for spring segment lengthening with $p<0.05$ noting statistical significance.

Muscularis thickness, crypt depth and villus length were similarly compared between control and spring-containing jejunum using paired t-test, with $p<0.05$ noting statistical significance.

## Results

Seven out of 8 pigs survived until POD 7. One pig was euthanized on POD 3 for failure to thrive from an unclear cause and was excluded from analysis. The remaining pigs did not have leak from the enterotomy site, intestinal perforation or bowel obstruction. The average weight at time of the operation was 9.3±1.2 kg, and at the time of euthanasia was 9.1±0.9kg, with an average 19.8 g weight loss per day.

### Springs

Springs were 7.5 cm in length prior to being compressed to 2 cm. All springs had a diameter of 10 mm. The average spring constant was 4.8±0.3 N/m and ranged from 4.2–5.0 N/m. The average initial spring force was 0.26±0.02 N. Springs deployed successfully with no evidence of perforation or obstruction. At euthanasia, two springs had slipped past proximal and distal plications, 2 springs had slipped distally only, 1 slipped proximally only, and 2 remained contained within both sets of plications.

### Jejunal lengthening

Segments of spring-containing jejunum measured an average of 2.5±0.4 cm in length initially, while control segments measured 2.2±0.4 cm. Spring-containing jejunum had a mean increase in length by 1.8±0.1 cm, whereas the control jejunum had decreased in length by a mean 0.2 ±0.1 cm. These changes in length corresponded to an absolute spring-segment increase in length by 72±36% and a relative lengthening by 89±39% when accounting for changes in control segment length (p = 0.004) (**Figs** 2 and 3).

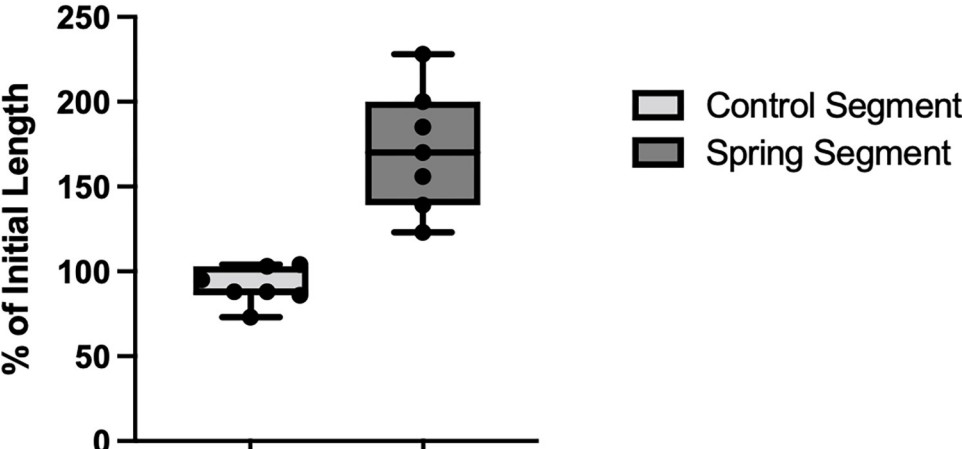

**Fig 3. Percentage change in length of spring-containing segment compared to control segment without spring.**
Expressed as % of initial length. Spring segments lengthened by 72±36% when compared to initial length and by 89
±39% when compared to controls (p = 0.004).

Only 2 springs remained within both sets of plications; these jejunal segments lengthened
by 99%, indicating a nearly two-fold lengthening effect. Springs that had slipped through the
plications lengthened the jejunum by an average of 61%.

## Histologic analysis

Lengthened jejunum maintained similar characteristics to control segments. Both mucosal
and muscular layers were intact on histologic analysis. There was no significant difference in
muscular thickness between control and spring segments (average 474±73 μm vs. 521±156 μm
respectively, p = 0.1). There was also no significant difference in muscularis thickness in spring
segments between slipped and affixed springs (p = 0.90). Crypt depth, however, was signifi-
cantly increased in spring-containing jejunum when compared to control jejunum (80±12 μm
vs. 64±11 μm respectively, p<0.001), as was villus length (978±104 μm vs. 661±84 μm,
p<0.001). (**Table** 1, **Fig** 4).

## Discussion

In this study, we sought to expand on our previous work on stimulating intestinal growth with
spring-mediated distraction by shifting from an external plication technique for spring fixa-
tion to an intraluminal approach with internal plication. We demonstrated that significant
intestinal lengthening is achieved when springs are implanted and fixed in place with this

**Table 1. Mean muscularis thickness, crypt depth and villus length in control and spring-containing porcine jejunum.**

|  | Control Segment (Mean ± St. Dev) | Spring Segment (Mean ± St. Dev) |
|---|---|---|
| Muscularis Thickness (μm) | 474±73 | 521±156 |
| Crypt Depth (μm) | 64±11 | 80±12 |
| Villus Length (μm) | 661±84 | 978±104 |

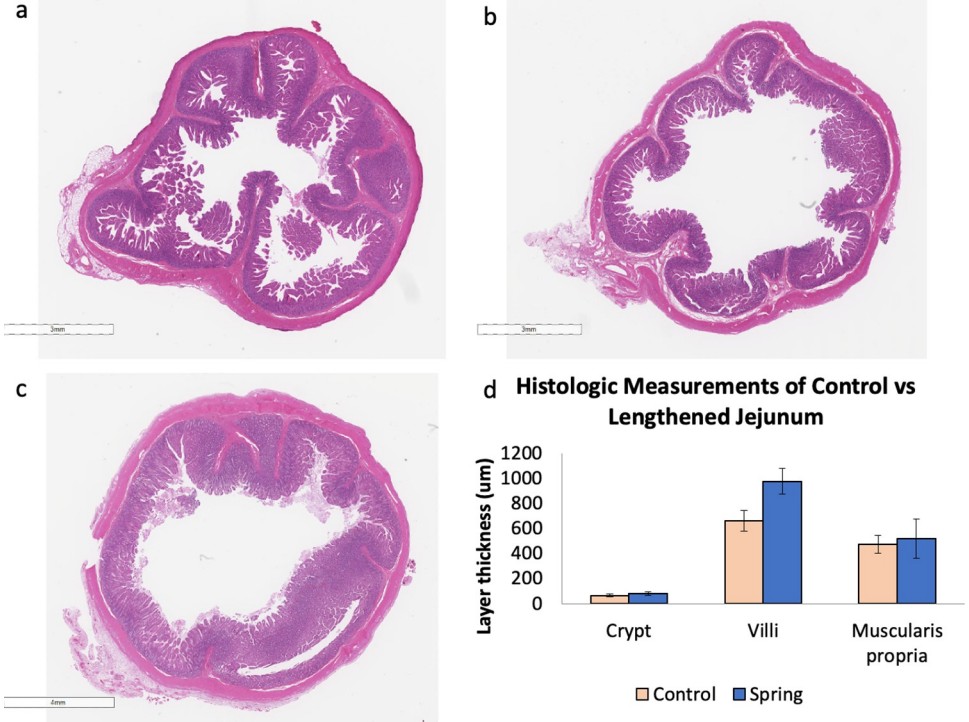

**Fig 4. Porcine jejunum after spring-mediated lengthening maintains intact mucosa and muscular layer.** (A) control jejunum, (B) internal plication site (C) spring segment after seven days in vivo. (D) Muscularis thickness, crypt depth and villus length in control and spring-lengthened jejunal segments.

technique. With evidence from this study showing the effectiveness of an intraluminal approach, we aim to further investigate less invasive methods of spring fixation and delivery.

Pigs did experience some weight loss throughout the 7-day postoperative course, but clinically appeared to be thriving and were able to maintain their liquid diet. There were no overt complications noted other than failure to thrive in one pig. The overall weight loss was likely attributed to the liquid diet and postoperative recovery.

We observed significant lengthening while utilizing the internal plication technique, similar to our findings with external plication [5–8,10,12,14]. We observed more spring slippage than we have in previous experiments with external plication. However, even with 5/7 slipped springs we observed a significant lengthening effect. This effect was blunted, however, with slipped springs demonstrating on average 38% less lengthening than springs that remained fixed within the plications. Thus this internal plication technique predisposes to spring slippage and may result in less consistent lengthening than our previously proven external plication technique. Springs that remained within the plications demonstrated the same two-fold lengthening effect that we have seen in previous experiments with external plication. Further experimentation is needed to perfect the internal plication technique to achieve consistent intestinal lengthening, particularly as the end goal is to be able to place these devices via an endoscopic approach [5,11].

Although a 1.7-fold increase in a single segment of intestinal length may appear modest, we have previously shown that multiple springs can be placed in continuity for an overall lengthening effect [5]. This implies that the number of springs that can be placed are limited only by existing bowel length. However, as we study this technique of mechanical distraction enterogenesis further, we look to re-lengthening previously lengthened intestine with reimplantation

of springs as a staged procedure. Additionally, we noted increased crypt depth and villus length in lengthened segments. This likely due to tissue generation as a result of a stimulating force being applied to the intestine as seen in our previous work [6,7,15,16].

Our ultimate goal is to develop a minimally invasive technique that precludes the need for laparotomy or enterotomy for spring delivery and fixation. We were able to show that we can achieve results similar to our previous experiments in porcine jejunum, even when accounting for spring slippage. With proof of effectiveness of spring-mediated intestinal lengthening via the internal plication technique, we demonstrated that an intraluminal approach was feasible even though it was performed using an open approach. The next step is to develop a completely endoscopic method for spring delivery with intraluminal fixation to produce consistent lengthening, thus limiting complications that accompany laparotomy and enterotomy. With an endoscopic approach, springs could be placed and confined luminally through a stoma or a natural orifice to achieve intestinal lengthening.

## Supporting information

**S1 Data.**
(XLSX)

## Acknowledgments

Eric Kramer from Stanford Byers Center for Biodesign for providing the furnace to manufacture the nitinol devices.

Dr. Samuel Baker and Greg Nelson from Stanford Veterinary Service Center for their help with animal procedures.

Eric Peterson and Doreen Wu from Stanford Animal Histology Services for their help with processing tissue samples for histology.

## Author Contributions

**Conceptualization:** James C. Y. Dunn.

**Data curation:** Talha A. Rafeeqi, Anne-Laure Thomas, Fereshteh Salimi-Jazi, James C. Y. Dunn.

**Formal analysis:** Talha A. Rafeeqi, Anne-Laure Thomas.

**Funding acquisition:** James C. Y. Dunn.

**Investigation:** Talha A. Rafeeqi, Anne-Laure Thomas, Fereshteh Salimi-Jazi, Modupeola Diyaolu.

**Methodology:** James C. Y. Dunn.

**Project administration:** Anne-Laure Thomas.

**Resources:** James C. Y. Dunn.

**Supervision:** James C. Y. Dunn.

**Validation:** James C. Y. Dunn.

**Writing – original draft:** Talha A. Rafeeqi.

**Writing – review & editing:** Talha A. Rafeeqi, Fereshteh Salimi-Jazi, James C. Y. Dunn.

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
