## [Decision Letter · Decision Letter 0]

14 Aug 2022

PONE-D-22-16047Internal plication for spring confinement to lengthen intestine in a porcine modelPLOS ONE

Dear Dr. Dunn,

Thank you for submitting your manuscript to PLOS ONE. After careful consideration, we feel that it has merit but does not fully meet PLOS ONE’s publication criteria as it currently stands. Therefore, we invite you to submit a revised version of the manuscript that addresses the points raised during the review process.

The decision is based on two reviewer reports which you can find at the end of this email. As you can see, both reviewers are supportive of publication, but minor changes to Fig 3 and 4 have been requested. 

We look forward to receiving your revised manuscript.

Kind regards,

Debora Walker

Editorial Office

PLOS ONE

Journal Requirements:

“This work was supported by the National Institutes of Health [R01DK130972].”

“This work was supported by the National Institutes of Health [R01DK130972] awarded to Dr. James Dunn. The NIH did not play a role in study design, data collection and analysis, decision to publish or preparation of the manuscript.

https://reporter.nih.gov/search/Lw0mL7cBJk-dGo_NlVYp_A/project-details/10338557”

“I have read the journal's policy and the authors of this manuscript have the following competing interests:

James CY Dunn, MD PhD is a co-founder of Eclipse Regenesis, a company that develops similar spring-based intestinal lengthening devices, and reported his patent on “Expandable distension device for hollow organ growth.”

Reviewers' comments:

Reviewer's Responses to Questions

**Comments to the Author**

1. Is the manuscript technically sound, and do the data support the conclusions?

Reviewer #1: Yes

Reviewer #2: Yes

2. Has the statistical analysis been performed appropriately and rigorously? 

Reviewer #1: Yes

Reviewer #2: Yes

3. Have the authors made all data underlying the findings in their manuscript fully available?

Reviewer #1: Yes

Reviewer #2: Yes

4. Is the manuscript presented in an intelligible fashion and written in standard English?

Reviewer #1: Yes

Reviewer #2: Yes

5. Review Comments to the Author

Reviewer #1: The Authors have investigated an interesting topic and the theme has been properly described. They sought to expand on their previous work on stimulating intestinal growth with spring-mediated distraction by shifting from an external plication technique for spring fixation to an intraluminal approach with internal plication. I would like to congratulate authors for the good-quality of the article, the literature reported used to write the paper, and for the clear and appropriate structure. The manuscript is well written, presented and discussed, and understandable to a specialist readership. So, I recommend the acceptance of this manuscript.

Minor points:

1. In figure 3, I suggest the authors change the position of control and treatment. In the view of readers, we want to know the control level and then use this level to compare with the treatment level.

2. In figure 4, I suggest the authors provide the crypt depth and muscularis propria thickness. Furthermore, it will be perfect if the authors could improve the quality of figure 4.

Reviewer #2: This research describes a treatment method for short bowel that is innovative and striking. Although the authors frankly describe that these are early and modest results in view of developing an endoscopic technique that is ambitious and complex, I encourage them to continue to advance this project and wish them future success for the benefit of this type of patient.

6. PLOS authors have the option to publish the peer review history of their article (what does this mean?). If published, this will include your full peer review and any attached files.

Reviewer #1: No

Reviewer #2: **Yes: **Cristians Gonzalez

---

## [Author Response · Author response to Decision Letter 0]

30 Aug 2022

1. We have edited the manuscript to comply with PLOS One’s style requirements

2. Table 1 has been added to the main text, on page 8. No additional supplementary tables are submitted. 

3. We have removed funding information in the manuscript and would not like to make any changes to the Funding Statement. 

4. We have removed competing interest information from the manuscript and would like to amend our Competing Interests Section with the following: 

“I have read the journal's policy and the authors of this manuscript have the following competing interests:

James CY Dunn, MD PhD is a co-founder of Eclipse Regenesis, a company that develops similar spring-based intestinal lengthening devices, and reported his patent on “Expandable distension device for hollow organ growth. This does not alter our adherence to PLOS ONE policies on sharing data and materials.”

5. There are no legal or ethical restrictions to sharing our data; as such, we have provided data pertinent to this study as a supplemental file of the name “S1_DATA.xlsx”

Reviewer #1: We appreciate your constructive remarks and have used these to improve our manuscript as follows. 1: Figure 3 has been amended as requested to show control segment length before spring segment length. 2: We have revised figure 4 to be better quality. Additionally, we have added a graph to figure 4 to show muscularis thickness, crypt depth and villus length.

Reviewer #2: Thank you for your kind words and your support as we move forward with this work.

---

## [Editor Report · Decision Letter 1]

1 Sep 2022

Internal plication for spring confinement to lengthen intestine in a porcine model

PONE-D-22-16047R1

Dear Dr. Dunn,

We’re pleased to inform you that your manuscript has been judged scientifically suitable for publication and will be formally accepted for publication once it meets all outstanding technical requirements.

Kind regards,

Emily Chenette

Editor in Chief

PLOS ONE
---

## [Editor Report · Acceptance letter]

6 Sep 2022

PONE-D-22-16047R1 

Internal plication for spring confinement to lengthen intestine in a porcine model 

Dear Dr. Dunn:

I'm pleased to inform you that your manuscript has been deemed suitable for publication in PLOS ONE. Congratulations! Your manuscript is now with our production department. 

Kind regards, 

on behalf of

Dr Emily Chenette 

Staff Editor

PLOS ONE